# An observational, multicenter, registry-based cohort study of Turkish Neonatal Society in neonates with Hypoxic ischemic encephalopathy

Emel Okulu[1]*, Ibrahim Murat Hirfanoglu[2], Mehmet Satar[3], Omer Erdeve[1], Esin Koc[2], Ferda Ozlu[3,4], Mahmut Gokce[4], Didem Armangil[5], Gaffari Tunc[6], Nihal Demirel[7,8], Sezin Unal[8], Ramazan Ozdemir[9], Mehmet Fatih Deveci[9], Melek Akar[10], Melike Kefeli Demirel[10], Merih Çetinkaya[11], Halime Sema Can Buker[11], Belma Saygılı Karagöl[12], Deniz Yaprak[12], Abdullah Barış Akcan[13], Ayse Anik[13], Fatma Narter[14], Sema Arayici[15], Egemen Yildirim[15], Ilke Mungan Akin[16], Ozlem Sahin[16], Ozgul Emel Bulut Ozdemir[17], Fahri Ovali[17], Mustafa Ali Akin[18], Yalcin Celik[19], Aysen Orman[19], Sinan Uslu[20], Hilal Ozkan[21], Nilgun Koksal[21], Ayhan Tastekin[22], Mehmet Gunduz[22], Ayse Engin Arisoy[23], Resat Gurpinar[23], Rahmi Ors[24], Huseyin Altunhan[25], Ramazan Kececi[25], Hacer Yapicioglu Yildizdas[3], Demet Terek[26], Mehmet Ates[26], Sebnem Kader[27], Mehmet Mutlu[27], Kıymet Çelik[28], Ebru Yucesoy[29], Mustafa Kurthan Mert[30], Selvi Gulasi[30], Kazım Kucuktasci[31], Didem Arman[32], Berna Hekimoglu[33], Nazlı Dilay Gultekin[34], Hasan Tolga Celik[35], Dilek Kahvecioglu[36], Can Akyildiz[37], Erdal Taşkın[38], Nukhet Aladag Ciftdemir[39], Saime Sundus Uygun[40], Tugba Barsan Kaya[41], Arzu Akdag[42], Aslan Yilmaz[43]

1 Ankara University Faculty of Medicine, Department of Pediatrics, Division of Neonatology, Ankara, Türkiye, 2 Gazi University Faculty of Medicine, Department of Pediatrics, Division of Neonatology, Ankara, Türkiye, 3 Cukurova University Faculty of Medicine, Department of Pediatrics, Division of Neonatology, Adana, Türkiye, 4 Department of Neonatology, Seyhan State Hospital, Adana, Türkiye, 5 Department of Pediatrics, Neonatal Intensive Care Unit, Koru Hospital, Ankara, Türkiye, 6 Cumhuriyet University Faculty of Medicine, Department of Pediatrics, Division of Neonatology, Sivas, Türkiye, 7 Yildirim Beyazit University Faculty of Medicine, Department of Pediatrics, Division of Neonatology, Ankara, Türkiye, 8 Department of Neonatology, Etlik Zubeyde Hanim Women's Health Teaching and Research Hospital, Ankara, Türkiye, 9 Inonu University Faculty of Medicine, Department of Pediatrics, Division of Neonatology, Malatya, Türkiye, 10 Department of Neonatology, University of Health Sciences, Tepecik Training and Research Hospital, Izmir Türkiye, 11 Department of Neonatology, University of Health Sciences, Basaksehir Çam and Sakura City Hospital, Istanbul, Türkiye, 12 Gulhane Faculty of Medicine, Department of Pediatrics, Division of Neonatology, University of Health Sciences, Ankara, Türkiye, 13 Adnan Menderes University Faculty of Medicine, Department of Pediatrics, Division of Neonatology, Aydin, Türkiye, 14 Department of Neonatology, University of Health Sciences, Kartal Dr. Lutfi Kirdar Education and Research Hospital, Istanbul, Türkiye, 15 Department of Neonatology, Eskisehir State Hospital, Eskisehir, Türkiye, 16 Department of Neonatology, University of Health Sciences, Umraniye Training and Research Hospital Istanbul, Istanbul, Türkiye, 17 Medeniyet University Faculty of Medicine, Department of Pediatrics, Division of Neonatology, Istanbul, Türkiye, 18 Ondokuz Mayis University Faculty of Medicine, Department of Pediatrics, Division of Neonatology, Samsun, Türkiye, 19 Mersin University Faculty of Medicine, Department of Pediatrics, Division of Neonatology, Mersin, Türkiye, 20 Department of Neonatology, University of Health Sciences, Sisli Etfal Hamidiye Training and Research Hospital, Istanbul, Türkiye, 21 Uludag University Faculty of Medicine, Department of Pediatrics, Division of Neonatology, Bursa, Türkiye, 22 Medipol University Faculty of Medicine, Department of Pediatrics, Division of Neonatology, Istanbul, Türkiye, 23 Kocaeli University Faculty of Medicine, Department of Pediatrics, Division of Neonatology, Kocaeli, Türkiye, 24 Department of Pediatrics, Neonatal Intensive Care Unit, Ozel Medova Hospital, Konya, Türkiye, 25 Necmettin Erbakan University Meram Faculty of Medicine, Department of Pediatrics, Division of Neonatology, Konya, Türkiye, 26 Ege University Faculty of Medicine, Department of Pediatrics, Division of Neonatology, Izmir, Türkiye, 27 Karadeniz Technical University Faculty of Medicine, Department of Pediatrics, Division of Neonatology, Trabzon, Türkiye, 28 Akdeniz University Faculty of Medicine, Department of Pediatrics, Division of Neonatology, Antalya, Türkiye, 29 Harran University Faculty of Medicine, Department of Pediatrics Division of Neonatology, Sanliurfa, Türkiye, 30 Department of Neonatology, University of Health Sciences, Adana, Türkiye, 31 Adana City Training and Research Hospital, Department of Pediatrics, Neonatal Intensive Care



**Data Availability Statement:** All relevant data are within the paper and its Supporting Information files.

**Funding:** The authors received no specific funding for this work.

**Competing interests:** The authors have declared that no competing interests exist.

Unit, Ozel Saglik Hospital, Denizli, Türkiye, **32** Department of Neonatology, Istanbul Training and Research Hospital, Istanbul, Türkiye, **33** Department of Neonatology, University of Health Sciences, Kanuni Training and Research Hospital, Trabzon, Türkiye, **34** Department of Neonatology, Van Regional Training and Research Hospital, Van, Türkiye, **35** Hacettepe University Faculty of Medicine, Department of Pediatrics, Division of Neonatology, Ankara, Türkiye, **36** Department of Neonatology, University of Health Sciences, Ankara Training and Research Hospital, Ankara, Türkiye, **37** Dokuz Eylul University Faculty of Medicine, Department of Pediatrics, Division of Neonatology, Izmir, Türkiye, **38** Fırat University Faculty of Medicine, Department of Pediatrics, Division of Neonatology, Elazığ, Türkiye, **39** Trakya University Faculty of Medicine, Department of Pediatrics, Division of Neonatology, Edirne, Türkiye, **40** Necmettin Erbakan University Selcuk Faculty of Medicine, Department of Pediatrics, Division of Neonatology, Konya, Türkiye, **41** Osmangazi University Faculty of Medicine, Department of Pediatrics, Division of Neonatology, Eskisehir, Türkiye, **42** Department of Neonatology, University of Health Sciences, Yuksek Ihtisas Teaching Hospital, Bursa, Türkiye, **43** Cerrahpasa University Faculty of Medicine, Department of Pediatrics, Division of Neonatology, Istanbul, Türkiye

* emelokulu@gmail.com

## Abstract

### Background

Hypoxic ischemic encephalopathy (HIE) is a significant cause of mortality and short- and long-term morbidities. Therapeutic hypothermia (TH) has been shown to be the standard care for HIE of infants $\geq$36 weeks gestational age (GA), as it has been demonstrated to reduce the rates of mortality, and adverse neurodevelopmental outcomes. This study aims to determine the incidence of HIE in our country, to assess the TH management in infants with HIE, and present short-term outcomes of these infants.

### Methods

The Turkish Hypoxic Ischemic Encephalopathy Online Registry database was established for this multicenter, prospective, observational, nationally-based cohort study to evaluate the data of infants born at $\geq$34 weeks GA who displayed evidence of neonatal encephalopathy (NE) between March, 2020 and April 2022.

### Results

The incidence of HIE among infants born at $\geq$36 weeks GA (n = 965) was 2.13 per 1000 live births (517:242440), and accounting for 1.55% (965:62062) of all neonatal intensive care unit admissions. The rates of mild, moderate and severe HİE were 25.5% (n = 246), 58.9% (n = 568), and 15.6% (n = 151), respectively. Infants with severe HIE had higher rates of abnormal magnetic resonance imaging (MRI) findings, and mortality (p<0.001). No significant difference in mortality and abnormal MRI results was found according to the time of TH initiation (<3 h, 3–6 h and >6 h) (p>0.05). TH was administered to 85 (34.5%) infants with mild HIE, and of those born of 34–35 weeks of GA, 67.4% (n = 31) received TH. A total of 58 (6%) deaths were reported with a higher mortality rate in infants born at 34–35 weeks of GA (OR 3.941, 95% CI 1.446–10.7422, p = 0.007).

### Conclusion

The incidence of HIE remained similar over time with a reduction in mortality rate. The timing of TH initiation, whether <3 or 3–6 h, did not result in lower occurrences of brain lesions on

MRI or mortality. An increasing number of infants with mild HIE and late preterm infants with HIE are receiving TH; however, the indications for TH require further clarification. Longer follow-up studies are necessary for this vulnerable population.

## Introduction

Hypoxic ischemic encephalopathy (HIE) is a major cause of mortality and short- and long-term morbidities. HIE is the subset of neonatal encephalopathy (NE) that is accepted to be caused by potentially asphyxiating birth events, or 'sentinel events' [1,2].

The estimated incidences of NE and HIE are 3/1000 and 1.5/1000 live births, respectively in developed countries, and it is thought to be higher in developing countries [3]. In a recent study from New Zealand, NE was found to have an incidence of 1.2/1000 live births, which is lower than the 2010 review's estimate [4]. However, there were significant differences in reported incidences between population-based and hospital-based research. It was estimated that the incidence range would have been 2-6/1000 for NE and 1-8/1000 live births for HIE [5]. According to the data published by the Hypoxic Ischemic Encephalopathy Study Group of the Turkish Neonatology Society in Türkiye in 2008, 93 infants out of 19,857 live births were diagnosed with HIE, resulting in a frequency of 2.6/1000 live births, and 1.2% among neonatal intensive care units (NICU) hospitalized patients [6].

Therapeutic hypothermia (TH) has been shown to be the standard treatment for HIE in infants with a gestational age (GA) of ≥36 weeks, and provides lower rates of mortality, cerebral palsy, hearing and visual impairment, and neurodevelopmental delay [7–9]. However, a recent study evaluating the use of TH for HIE in low-income countries found it to be neither effective nor safe, and advises against its use [10], in which these results and conclusions should be carefully considered. There is not enough evidence to determine any significant benefits or harms from using TH in infants with mild HIE [11,12].

This study aimed to determine the incidence of HIE in Türkiye, assess TH management in infants with HIE, and present short-term outcomes of these infants, including the underlying etiologies, clinical features, morbidity and mortality.

## Material and methods

This was a multicenter, prospective, observational, nationally-based cohort study conducted between 15 March, 2020 and 15 April 2022. After the Turkish Hypoxic Ischemic Encephalopathy Online Registry was established, clinical directors in 64 NICUs nationwide were made aware of the study. The study included 42 (66%) participating tertiary level NICU centers among which 24 were university hospitals, 15 were state hospitals and 3 were private hospitals. All participating centers all had an attending neonatologists and almost all included a pediatric neurologists.

Infants who met the following criteria and were born at or transferred to a participating center at ≥34 weeks GA with evidence of HIE were eligible for enrollment in the registry [6,13]:

1. One or more of the following:

    i. Apgar score <5 at 5 min,

    ii. Metabolic acidosis [Base deficit (BD) > –16 mmol/L at cord blood gas or blood gas analysis at 1st h after birth),

 iii. Delayed onset of respiration for five or more minutes,

 iv. Birth via emergency cesarean section due to fetal distress

2. Need for ventilation immediately after birth (positive pressure ventilation or intubation)

3. Evidence of encephalopathy (lethargy, hypotonia, altered state of consciousness, weakness/ absent of reflexes and/or seizures)

4. Multiorgan involvement [encephalopathy and at least one other organ system other than central nervous system (CNS)].

For multiorgan dysfunction, the following conditions were considered [14–16]:

- Renal: Oliguria/anuria, hematuria, proteinuria, myoglobunuria, and renal failure

- Cardiovascular: Hypotension treated with an inotrope for more than 24 hours to maintain blood pressure within the normal range, shock, cardiomegaly, arrythmia, heart failure, cardiac ischemia

- Metabolic: Hypo/hyperglycemia, hypocalcemia, hyponatremia, hypomagnesemia, metabolic acidosis

- Pulmonary: Need for ventilator support with oxygen requirement, requirement of inhaled nitric oxide, need for extracorporeal membrane oxygenation

- Hepatic: Aspartate aminotransferase >100 IU/l or alanine aminotransferase >100 IU/l at any time during the first week after birth

- Hematological: Thrombocytopenia, thrombosis, disseminated intravascular coagulation, impaired coagulation profile

- Gastrointestinal: Bleeding, necrotizing enterocolitis

- Skin: Skin injuries that were considered device related when they were in contact with a cooling blanket, and not present at the of hospital admission.

The participating centers were asked to report factors identified as the underlying etiology: antepartum [maternal/fetal: advanced maternal age, maternal disease (hypertension, diabetes, thyroid, cardiac, preeclampsia/eclampsia, anemia, infection), maternal substance use, pre/ postmaturity, intrauterine growth restriction, multiple gestation, congenital malformation of fetus], intrapartum [uterine rupture, placental abruption, placenta previa, cord prolapse, tight nuchal cord, inflammatory events (maternal fever, chorioamnionitis, prolonged rupture of membranes), meconium-stained amniotic fluid], and postpartum (meconium aspiration syndrome, severe cardiopulmonary failure).

Encephalopathy was defined as the presence of moderate or severe encephalopathy ≥1 sign in at least 3 of the 6 categories of modified Sarnat criteria: 1. level of consciousness: moderate (lethargy), severe (stupor or coma), 2. spontaneous activity: moderate (decreased), severe (no activity), 3. posture: moderate (distal flexion, complete extension), severe (decerebrate), 4. tone: moderate (hypotonia), severe (flaccid), 5. primitive reflexes, suck: moderate (weak), severe (absent) or Moro: moderate (incomplete), severe (absent) and 6. autonomic nervous system, pupils: moderate (constricted), severe (deviated, dilated or nonreaction to light) or heart rate: moderate (bradycardia), severe (variable) or respiration: moderate (periodic breathing), severe (apnea). The number of moderate or severe signs determined the degree of encephalopathy; if signs were distributed equally, then the designation was based on the level of consciousness [17,18].

To determine the incidence of HIE, the number of infants admitted to the participating units during the study period, whether the cases included in the study group were born inborn or outborn, and the number of deliveries performed in the same center during the study period was asked from the participating centers.

Data for each neonate was recorded, including GA, birth weight (BW), sex, place of birth, APGAR scores at 5-, 10-, 15- and 20-min, mode of delivery, delivery room resuscitation, underlying etiologies, Thompson scores at admission and during the first 72 h after birth [19], and whether cooling therapy was given. Care practices, such as amplitude electroencephalography (aEEG) at admission, timing of nactive TH initiation with servo-controlled device after birth (in h), the duration of TH, need for respiratory support, accompanying organ systems issues aside from CNS condition, the results of MRI, EEG, and hearing screening test, and mortality rate were all recorded. Mortality was defined as mortality before NICU discharge. Infants were grouped into three groups according to TH initiation time as <3 h, 3–6 h and >6 h. Initiation of TH within <3 h was defined as 'early' initiation.

The aEEG background activity was classified as follows [20]:

1. Normal amplitude, the upper margin of band of aEEG activity >10 µV and the lower margin >5 µV;

2. Moderately abnormal amplitude, the upper margin of band of aEEG activity >10 µV and the lower margin ≤5 µV; and

3. Suppressed amplitude, the upper margin of the band of aEEG activity <10 µV and lower margin <5 µV, usually accompanied by bursts of high-voltage activity ("burst suppression")

The findings shown by MRI and defined as 'abnormal' were as follows [21]:

- Major conventional MRI findings: Cerebral cortical gray-white differentiation lost (on T1W or T2W); Cerebral cortical high signal (T1W and FLAIR), especially in parasagittal perirolandic cortex; Basal ganglia/thalamus, high signal (T1W and FLAIR, usually associated with the cerebral cortical changes but possibly alone with increased signal in brain stem tegmentum in cases of acute severe insults; Parasagittal cerebral cortex, subcortical white matter, high signal (T1W and FLAIR); Periventricular white matter, decreased signal (T1W) or increased signal (T2W); Posterior limb of internal capsule, decreased signal (T1W or FLAIR); Cerebrum in a vascular distribution,

- Diffusion-weighted MRI: Decreased diffusion (increased signal) in injured areas.

Only de-identified data were provided to the registry, negating the need for informed consent. The local institutional review board reviewed and approved registry participation with the approval No. of 25.11.2019–227. Authors had access to information that belongs to only their own center's participants during or after data collection.

Categorical variables were reported as number and percentage, and continuous variables as mean ± SD or median (IQR) where appropriate. Chi-square and Fisher's exact tests were used for categorical variables. We used the Student-t test to analyze continuous variables with a normal distribution, and the Mann-Whitney U test for non-normally distributed data. For group analysis, the One Way ANOVA test was used for normal distributed data, and the Kruskal-Wallis H test for non-normally distributed ones. We used the post hoc Bonferroni test to analyze the differences between each group. Logistic regression was applied to calculate the odss ratio to access association between risk and covariates. The optimum cutoff of the Thompson scores for estimating the mortality was determined using Receiver Operating Characteristic

(ROC) curves and area under the curve (AUC). Logistic regression P values of <0.05 were considered statistically significant. All statistical analyses were performed using SPSS version 11.5.

## Results

### Incidence of HIE

Among registered 1011 infants with HIE, 46 infants born <36 weeks of GA were excluded when calculating the incidence of HIE. Participating hospitals recorded 242440 births during the study period, and a total of 62062 infants were admitted for neonatal care. Among them, 53.6% (n = 517) were inborn, 45.9% (n = 443) were outborn, and 0.5% (n = 5) were born at home. The incidence of HIE in infants who are ≥36 weeks GA was 2.13 per 1000 live births (517:242440), and accounting for 1.55% (965:62062) of all NICU admissions. The incidence for mild HIE was 0.64 per 1000 births and 0.39%, and for moderate/severe HIE, it was 1.49 per 1000 births and 1.16%, respectively.

### The characteristics of infants

Of the total 965 infants, 25.5% (n = 246) had mild, 58.9% (n = 568), had moderate, and 15.6% (n = 151) had severe HIE (Fig 1). The mean GA and BW for these infants were 38.9 ± 1.4 w, and 3219 ± 504 g, respectively. The mean maternal age was 28.0 ± 5.9 years, 57.7% of the infants were male, and 57.3% born vaginally. Most of the infants received resuscitation in the delivery room (83.3%). The median APGAR scores at 1-, 5-, 10-, 15- and 20-min were 4, 6, 7, 7, and 7, respectively (Table 1).

### Data of infants ≥36 weeks GA according to grading of HIE

The characteristics of infants ≥36 weeks GA with mild, moderate and severe HIE were compared on Tables 1 and 2. The GA, BW, sex distribution, and maternal age were similar among infants (p>0.05). Antepartum conditions (maternal/fetal) were responsible for 24.8%, intrapartum conditions for 41.2%, and postpartum conditions in 10.3% of infants as underlying etiologies, while the etiology was unknown in 23.7% of infants with HIE (p = 0.195). Among the identified etiologies, diabetes (30%) and hypertension (22.8%) were the most common antepartum maternal problems, intrauterine growth restriction (14.6%) was the most common antepartum fetal problem; delivery through meconium-stained amniotic fluid (43.5%) and umbilical cord compression (19.3%) were the most common intrapartum problems; respiratory disease due to meconium aspiration syndrome (63%) was the most common postpartum problem (Table 1).

The place of birth, the APGAR scores at 1-, 5-, 10-, 15-, and 20-min and the rate delivery room resuscitation were different between groups according to degree of HIE (p<0.05) (p<0.001) (Table 1).

The clinical and laboratory findings, as well as short-term outcomes for infants are presented in Table 2. Significant differences between groups were observed in the pH and BE in the cord and first hour blood gas analysis, and the rate of abnormal aEEG at admission (p<0.001). All NICUs administered whole body servo controlled TH, and performed TH to 100% of severe HIE, to 99.1% of moderate HIE, and to 34.6% of mild HIE neonates. The Thompson scores at admission, at 24th, 48th and 72nd h (overall with medians of 16, 14, 12 and 9, respectively) were different according to severity of HIE (p<0.001) (Table 2).

The frequency of clinical seizures observed at any time during hospitalization varied among groups (p<0.001), with phenobarbital being the most commonly used first-line agent. 65.3%

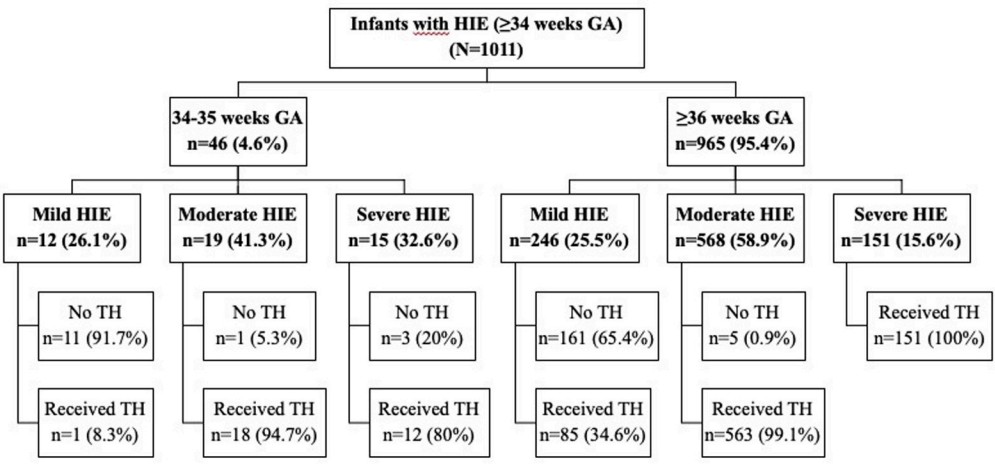

GA: gestational age; HIE: hypoxic ischemic encephalopathy; TH: therapeutic hypothermia

**Fig 1. The flowchart of the study.**

of all infants with any grade of HIE had at least one accompanying organ system or condition during hospitalization. The most frequent problems were respiratory problems, inotrope required hypotension and electrolyte imbalances. During their hospitalization, almost all infants with severe HIE (90.7%) had at least one problem, while 71.8% of moderate and 34.6% of mild HIE infants had at least one problem (p<0.001) (Table 2).

Although conventional EEG was not performed to all infants, EEG findings were abnormal in 63.5%, 28.7%, and 15.6% of infants with severe, moderate, and mild HIE, respectively (p<0.001). MRI was performed to 62.9% of severe HIE infants with 68.4% of abnormal results at a median age of 7 days, to 75.5% of moderate HIE infants with 29.1% of abnormal results at a median age of 5 days, and to 45.5% of mild HIE infants with 22.3% of abnormal results at a median age of 5 days (p<0.001) (Table 2).

None of the infants with mild HIE died. The rate of mortality was the highest in severe HIE neonates (29.8%) (p<0.001). The duration of hospital stay was different among groups (severe HIE: median 15 days; moderate HIE: median 11 days; mild HIE: median 8 days) (p<0.001) (Table 2).

## Timing of hypothermia in moderate/severe HIE

Among 719 infants with moderate/severe HIE, only 5 infants with moderate HIE did not receive TH. The median TH initiation time was 2 h (IQR: 1–4 h) for both moderate and severe HIE who received TH (p = 0.589). When these infants were grouped according to TH initiation time, TH was started <3 h, 3–6 h and >6 h in 55.5%, 41.4%, and 3.1% of infants, respectively. The median initiation time was 9.5 h (IQR: 7.375–12.25) for infants who received TH >6 h. The distribution of infants according to degree of HIE was similar among TH initiation timing groups (p = 0.320). TH initiated <3 h in 85.1% of inborn infants, and 75% of home-delivered infants, whereas TH initiated 3–6 h in 68.6% of outborn infants (p<0.001).

The rates of clinical seizure observed at any time and at least one accompanying organ system or condition during hospitalization were similar in TH initiation time groups (p = 0.829, and p = 0.492, respectively). There were no significant differences for abnormal conventional EEG findings, abnormal MRI findings and mortality and abnormal MRI results between the groups according to TH initiation time (p = 0.061, p = 0.348, and p = 0.809, respectively).

**Table 1. Demographic findings of infants with the diagnosis of HIE born ≥36 weeks of gestational age.**

| Variables | Total (N = 965) | HIE stages based on Modified Sarnat Scoring System | | | p |
|---|---|---|---|---|---|
| | | Mild HIE (Stage 1) (n = 246) | Moderate HIE (Stage 2) (n = 568) | Severe HIE (Stage 3) (n = 151) | |
| Gestational age (w)[*] | 38.9 ± 1.4 | 38.9 ± 1.4 | 38.8 ± 1.4 | 38.7 ± 1.3 | 0.378[w] |
| Birth weight (g)[*] | 3219 ± 504 | 3256 ± 512 | 3206 ± 479 | 3210 ± 584 | 0.413[w] |
| Maternal age (y)[*] | 28.0 ± 5.9 | 27.3 ± 5.9 | 28.3 ± 5.9 | 28.3 ± 6.0 | 0.075[w] |
| Sex (male)[‡] | 557 (57.7) | 135 (54.9) | 333 (58.6) | 89 (58.9) | 0.578[y] |
| **Place of birth**[‡] | | | | | <0.001[z] |
| *Inborn* | 517 (53.6) | 156 (63.4)[a] | 289 (50.9)[b] | 72 (47.7)[b] | |
| *Outborn* | 443 (45.9) | 89 (36.2)[a] | 278 (48.9)[b] | 76 (50.3)[b] | |
| *Home* | 5 (0.5) | 1 (0.4)[a,b] | 1 (0.2)[b] | 3 (2)[a] | |
| **Type of delivery (NVD)**[‡] | 553 (57.3) | 142 (57.7)[a,b] | 341 (60)[a] | 70 (46.4)[b] | 0.01[y] |
| **APGAR scores**[š] | | | | | |
| *1ˢᵗ min (n = 965)* | 4 (1–2) | 5 (4–6)[a] | 4 (3–5)[b] | 1 (1–3)[c] | <0.001[x] |
| *5ᵗʰ min (n = 965)* | 6 (5–6) | 7 (6–8)[a] | 6 (5–7)[b] | 4 (3–5)[c] | <0.001[x] |
| *10ᵗʰ min (n = 537)* | 7 (5–8) | 8 (7–9)[a] | 7 (6–8)[b] | 5 (4–6)[c] | <0.001[x] |
| *15ᵗʰ min (n = 236)* | 7 (6–8) | 8 (8–10)[a] | 7 (6–8)[b] | 6 (5–7)[c] | <0.001[x] |
| *20ᵗʰ min (n = 207)* | 7 (7–9) | 9 (8–10)[a] | 8 (7–9)[b] | 7 (5–7.5)[c] | <0.001[x] |
| **Resuscitation at DR**[‡] | 804 (83.3) | 176 (71.5)[a] | 483 (85)[b] | 145 (96)[c] | <0.001[y] |
| **Duration of resuscitation** (n = 804)(min)[š] | 2 (1–5) | 1 (1–2)[a] | 2 (1–3)[b] | 5 (2–10)[c] | <0.001[x] |
| **Underlying etiologies**[‡] | | | | | 0.195[y] |
| *Antepartum (maternal/fetal)* | 239 (24.8) | 66 (26.8) | 135 (23.8) | 38 (25.2) | 0.644[y] |
| Maternal diabetes | 42/140 (30) | 15/42 (35.7) | 23/75 (30.7) | 4/23 (17.4) | 0.084[y] |
| Maternal hypertension | 13/140 (9.3) | 3/42 (7.1) | 10/75 (13.3) | 0/23 (0) | 0.021[y] |
| Intrauterine growth restriction | 18/123 (14.6) | 4/29 (13.8) | 12/73 (16.4) | 2/21 (9.5) | |
| *Intrapartum* | 398 (41.2) | 87 (35.4) | 243 (42.8) | 68 (45) | |
| Meconium-stained amniotic fluid | 173 (43.5) | 41 (47.1) | 100 (41.1) | 32 (47) | |
| Umbilical cord compression | 77 (19.3) | 14 (16.1) | 50 (20.6) | 13 (19.1) | |
| *Postpartum* | 99 (10.3) | 22 (8.9)[a,b] | 52 (9.2)[a] | 25 (16.6)[b] | |
| Meconium aspiration syndrome | 63 (63.6) | 16 (72.7) | 33 (63.5) | 14 (56) | |
| *Unknown* | 229 (23.7) | 71 (28.9) | 138 (24.3) | 20 (13.2) | |

Data given as: [*]mean ± SD, [‡]number (%), [š]median (IQR). *w:One Way ANOVA test*, *x:Kruskal Wallis H test*, *y:Chi-square test*, *z:Fisher-exact test*.

Each subscript [a,b,c] letter denotes a subset of HIE categories whose column proprotions do not significantly from each other at the 0.05 level..

DR: delivery room; HIE: hypoxic ischemic encephalopathy; NVD: normal vaginal delivery; PPV: positive pressure ventilation.

Logistic regression analyses using TH initiation time (<3 h, 3–6 h, and >6 h) adjusting the severity of HIE and place of birth revealed no significant differences between groups for both mortality and abnormal MRI findings (Table 3).

In Kaplan–Meier survival analysis timing of TH was not associated with mortality (p = 0.784) (Fig 2). When the initiation time for TH within 6 h was compared as <3 h and 3–6 h, there were also no significant differences for mortality and abnormal MRI results (p = 0.651, and p = 0.154, respectively).

## Infants ≥36 weeks with mild HIE

There were 246 infants with mild HIE born at ≥36 weeks GA. The characteristics of infants who received TH (n = 85) and who did not (n = 161) were compared (Table 4).

The mean BW of infants who received TH was lower than who did not (p = 0.021). Most of infants who did not received TH (75.2%) were inborn, whereas 58.8% of infants who received TH were outborn (p<0.001). The type of delivery, need for resuscitation in the delivery room, and underlying etiologies were similar (p = 0.426, and p = 0,724, respectively). Infants who received TH had lower APGAR scores at 1ˢᵗ min (p = 0.013). The cord blood gas analyses were

**Table 2. Clinical features and outcomes of infants according to degree of HIE.**

| Variables | Mild HIE (n = 246) | Moderate HIE (n = 568) | Severe HIE (n = 151) | p |
|---|---|---|---|---|
| **Cord blood gas analysis** | | | | |
| pH$^\text{š}$ | 7.00 (6.93–7.07)[a] | 6.97 (6.89–7.02)[b] | 6.85 (6.77–7.00)[c] | <0.001[w] |
| BE$^\text{š}$ | -12.77 [(-15.57)–(-12.00)][a] | -17.00 [(-20.00)–(-14.00)][b] | -19.30 [(-23.50)–(-16.80)][c] | <0.001[x] |
| **Blood gas analysis at 1$^\text{st}$ h** | | | | |
| pH$^\text{š}$ | 7.14 (7.05–7.25)[a] | 7.07 (7.00–7.17)[b] | 6.93 (6.80–7.06)[c] | <0.001[w] |
| BE$^\text{š}$ | -10.41 [(-14.00)–(-6.92)][a] | -14.39 [(-17.30)–(-11.00)][b] | -19.00 [(-22.60)–(-14.90)][c] | <0.001[x] |
| **Abnormal aEEG at admission$^\ddagger$** | 26/211 (12.3)[a] | 269/511 (52.6)[b] | 119/137 (86.8)[c] | <0.001[y] |
| **Performed TH** | 85 (34.6)[a] | 563 (99.1)[b] | 151 (100)[b] | <0.001[y] |
| **Timing initiation of TH (h)$^\text{š}$** | 3.0 (1.0–4.75) | 2.0 (1.0–4.0) | 2.0 (1.0–4.0) | 0.144[x] |
| **Respiratory support during TH$^\ddagger$** | 160 (65.0)[a] | 480 (84.5)[b] | 150 (99.3)[c] | <0.001[y] |
| **Clinical seizure at any time$^\ddagger$** | 9 (3.7)[a] | 170 (29.9)[b] | 91 (60.3)[c] | <0.001[y] |
| **Accompanying organ system/condition$^\ddagger$** | 85 (34.6)[a] | 408 (71.8)[b] | 137 (90.7)[c] | <0.001[y] |
| Intractable acidosis | 2 (0.8)[a] | 34 (6.0)[b] | 56 (37.1)[c] | <0.001[y] |
| Received inotrope/hypotension | 14 (5.7)[a] | 148 (26.1)[b] | 91 (60.3)[c] | <0.001[y] |
| Bleeding/DIC | 8 (3.3)[a] | 78 (13.7)[b] | 68 (45.0)[c] | <0.001[y] |
| Thrombocytopenia | 22 (8.9)[a] | 115 (20.2)[b] | 72 (47.7)[c] | <0.001[y] |
| Blood glucose disturbance | 18 (7.3)[a] | 65 (11.4)[a] | 37 (24.5)[b] | <0.001[y] |
| Electrolyte imbalance | 25 (10.2)[a] | 149 (26.2)[b] | 77 (51.0)[c] | <0.001[y] |
| NEC | 0 (0)[a] | 23 (4.0)[b] | 2 (1.3)[a,b] | 0.002[y] |
| Acute kidney injury | 12 (4.9)[a] | 41 (7.2)[a] | 56 (37.1)[b] | <0.001[y] |
| Respiratory problems | 23 (9.3)[a] | 163 (28.7)[b] | 75 (49.7)[c] | <0.001[y] |
| Received iNO | 3 (1.2)[a] | 17 (3.0)[a,b] | 10 (6.6)[b] | 0.01[y] |
| Liver dysfunction | 15 (6.1)[a] | 111 (19.5)[b] | 74 (49.0)[c] | <0.001[y] |
| Received ECMO | 1 (0.4) | 0 (0.0) | 1 (0.7) | 0.206[z] |
| Skin problems | 2 (0.8) | 14 (2.5) | 6 (4.0) | 0.110[y] |
| **Thompson scores$^\text{š}$** | | | | |
| At admission | 4.0 (2.0–6.0)[a] | 9.0 (7.0–12.0)[b] | 16.0 (13.0–18.0)[c] | <0.001[x] |
| 24$^\text{th}$ h | 1.0 (0.0–3.0)[a] | 7.0 (3.0–10.0)[b] | 14.0 (11.0–17.0)[c] | <0.001[x] |
| 48$^\text{th}$ h | 0.0 (0.0–1.0)[a] | 4.0 (2.0–8.0)[b] | 12.0 (8.0–15.0)[c] | <0.001[x] |
| 72$^\text{nd}$ h | 0.0 (0.0–0.0)[a] | 2.0 (0.0–6.0)[b] | 9.0 (5.0–14.0)[c] | <0.001[x] |
| **Abnormal EEG finding$^\ddagger$** | 12/77 (15.6)[a] | 94/327 (28.7)[a] | 40/63 (63.5)[b] | <0.001[y] |
| **Performed MRI$^\ddagger$** | 112 (45.5)[a] | 429 (75.5)[b] | 95 (62.9)[c] | <0.001[y] |
| **Type of MRI$^\ddagger$** | 28 (25.0) | 74 (17.2) | 14 (14.7) | 0.286[z] |
| Conventional | 83 (74.1) | 348 (81.1) | 79 (83.2) | <0.001[x] |
| DWI | 1 (0.9) | 7 (1.6) | 2 (2.1) | <0.001[y] |
| Spectroscopy | 5.0 (4.0–7.75) | 5.0 (5.0–7.0) | 7.0 (5.0–13.0) | |
| **Timing of MRI (d)$^\text{š}$** | 25 (22.3)[a] | 125 (29.1)[a] | 65 (68.4)[b] | |
| **Abnormal MRI finding$^\ddagger$** | | | | |
| **Outcomes (mortality)$^\ddagger$** | 0 (0.0)[a] | 13 (2.3)[a] | 45 (29.8)[b] | <0.001[y] |
| Age at mortality$^\text{š}$ | - | 4.0 (3.0–5.5) | 3.0 (1.5–10.0) | 0.720[v] |
| **Length of hospital stay$^\text{š}$** | 8.0 (6.0–12.0)[a] | 11.0 (8.0–15.0)[b] | 15.0 (9.0–27.0)[c] | <0.001[x] |

Data given as: *mean ± SD, $^\ddagger$number (%), $^\text{š}$median (IQR); w:*One Way ANOVA test*, x:*Kruskal Wallis H test*, y:*Chi-square test*, z:*Fisher-exact test*; v: *Mann-Whitney test* (for mortality in moderate and severe HIE groups).

Each subscript [a,b,c] letter denotes a subset of HIE categories whose column proprotions do not significantly from each other at the 0.05 level.

aEEG: amplitude electroencephalography DWI: diffusion-weighted; ECMO: extracorporeal membrane oxygenation; EEG: electroencephalography; HIE: hypoxic ischemic encephalopathy; iNO: inhaled nitric oxide; MRI: magnetic resonance imaging; NEC: necrotizing enterocolitis; TH: therapeutic hypothermia.

different in groups (for pH and BE, p = 0.001 and p = 0.008, respectively). Thompson scores at admission, 24$^\text{th}$, 48$^\text{th}$, and 72$^\text{nd}$ h were all higher in infants who received TH (p = 0.029, p<0.001, p<0.001 and p<0.001, respectively). Infants who received TH had more problems

**Table 3. Mortality and abnormal MRI findings according to TH initiation time.**

|  | n (%) | OR (95% CI) | p | Adjusted OR* (95% CI) | p |
|---|---|---|---|---|---|
| **Mortality** | | | | | |
| **<3 h (n = 396)** | 31 (7.8) | - | | | |
| **3–6 h (n = 296)** | 26 (8.8) | 1.134 (0.658–1.954) | 0.651 | 1.916 (0.869–4.222) | 0.107 |
| **>6 h (n = 22)** | 1 (4.5) | 0.561 (0.073–4.309) | 0.578 | 2.047 (0.208–20.133) | 0.539 |
| **Abnormal MRI finding** | | | | | |
| **<3 h (n = 277)** | 93 (33.6) | - | | | |
| **3–6 h (n = 224)** | 89 (39.7) | 1.304 (0.905–1.880) | 0.155 | 0.894 (0.559–1.429) | 0.639 |
| **>6 h (n = 21)** | 7 (33.3) | 0.989 (0.386–2.535) | 0.982 | 0.750 (0.269–2.087) | 0.581 |

*Adjusted for severity of HIE and place of birth.

MRI: magnetic resonance imaging.

than who did not (p = 0.001). MRI and EEG were performed for more infants who received TH (MRI: 68.2% vs. 33.5%, p<0.001; EEG: 62.4% vs. 14.9%, p<0.001), but the rate of abnormal MRI and EEG findings were similar (p = 0.632, and p = 0.175). The duration of hospital stay was longer in infants who received TH (p<0.001) (Table 4).

## Infants born at 34–35 weeks

In total, 46 infants at 34 and 35 weeks were diagnosed with any grade HIE. Of these, 26.1% (n = 12) had mild, 41.3% (n = 19) had moderate, and 32.6% (n = 15) had severe HIE. Among these infants, 67.4% received TH, of which almost all were with moderate/severe HIE except one infant. APGAR scores and underlying etiologies were similar between the infants who received TH and who did not (p>0.05). Thompson scores were higher in infants who received TH (p<0.05). The rate of mortality was similar according TH application (p = 0.132) (Table 5).

Overall, the rate of mortality in infants with moderate/severe HIE born at 34–35 weeks of GA was 21.7% (n = 10) compared with those born ≥36 weeks of GA at 6% (p<0.001). Logistic

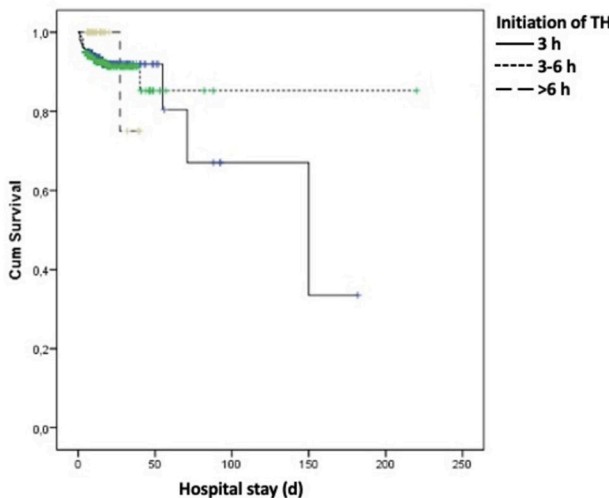

**Fig 2. Survival analysis and interpretation of initiation time of TH.**

**Table 4. Characteristics of infants with mild HIE according to TH application.**

| Variables | No TH (n = 161) | Received TH (n = 85) | p |
|---|---|---|---|
| Gestational age (w)[*] | 38.9 ± 1.4 | 38.7 ± 1.4 | 0.246[w] |
| Birth weight (g)[*] | 3311 ± 493 | 3153 ± 533 | 0.021[w] |
| Place of birth[‡] | | | <0.001[z] |
| *Inborn* | 121 (75.2) | 35 (41.2) | |
| *Outborn* | 39 (24.2) | 50 (58.8) | |
| *Home* | 1 (0.6) | 0 (0) | |
| APGAR scores[š] | | | |
| *1st min* | 5 (4–6) | 4 (3–6) | 0.013[x] |
| *5th min* | 7 (6–8) | 7 (6–8) | 0.464[x] |
| *10th min* | 8 (7–9) | 8 (7–8) | 0.503[x] |
| Cord blood gas analysis | n = 96 | n = 50 | |
| *pH[š]* | 7.02 (6.98–7.08) | 6.98 (6.90–7.02) | 0.001[w] |
| *BE[š]* | -13.1 [(-14.5)–(-11.62)] | -15.00 [(-17.15)–(-12.00)] | 0.008[x] |
| Blood gas analysis at 1st h | n = 160 | n = 80 | |
| *pH[š]* | 7.15 (7.07–7.23) | 7.12 (7.03–7.26) | 0.286[w] |
| *BE[š]* | -11.0 [(-13.45)–(-6.72)] | -12.0 [(-15.00)–(-7.62)] | 0.154[x] |
| Thompson scores[š] | | | |
| *At admission* | 3.0 (2.0–5.0) | 5.0 (2.0–7.0) | 0.029[x] |
| *24th h* | 0.0 (0.0–2.0) | 3.0 (0.0–4.0) | <0.001[x] |
| *48th h* | 0.0 (0.0–1.0) | 1.0 (0.0–3.0) | <0.001[x] |
| *72nd h* | 0.0 (0.0–0.0) | 0.0 (0.0–1.0) | <0.001[x] |
| Abnormal EEG finding[‡] | 6/53 (11.3) | 6/24 (25) | 0.175[y] |
| Abnormal MRI finding[‡] | 11/54 (20.4) | 14/58 (24.1) | 0.632[y] |
| Length of hospital stay (d)[š] | 7.0 (6.0–10.0) | 10.0 (8.0–14.0) | <0.001[x] |

Data given as: [*] mean ± SD, [‡] number (%), [š] median (IQR).

EEG: electroencephalography; MRI: magnetic resonance imaging.

w:Student-t test, x:Mann Whitney U test, y:Chi-square test,z:Fisher-exact test.

regression analysis adjusting receiving TH, and severity of HIE showed also difference in mortality according to GA (OR 3.941, 95% Cl 1.446–10.7422, p = 0.007).

## In-hospital mortality

A total of 58 (6%) deaths was reported, comprising 2.3% of the moderate, and 29.8% of the severe HIE neonates born at ≥36 weeks GA (p<0.001) of which death was occurred in similar days (Tables 1 and 2). The mean GA of non-survivor infants was lower accompanying higher need for delivery room resuscitation and lower APGAR scores (p<0.05) (Table 6).

The Thompson scores at admission, 24th h, 48th h, and 72nd h showed AUCs of 0.785 (95% Cl 0.723–0.847), 0.838 (95% Cl 0.777–0.898), 0.839 (95% Cl 0.766–0.913), and 0.864 (95%Cl 0.793–0.935) for predicting mortality, respectively. The Thompson scores of >10 at admission, 24th h, and 48th h showed that it had sensitivity and specificity of 0.810 and 0.546, 0.793 and 0.732, 0.719, and 0.85, respectively for mortality. The Thompson score of >8 at 72nd h showed a sensitivity of 0.842 and a specificity of 0.87.

All of the non-survivor infants received TH, but there was no significant difference in the timing of TH between groups (p = 0.794). Non-survivor infants were significantly more likely to have clinical seizures or any problem during hospitalization, abnormal EEG, and MRI findings (Table 6).

**Table 5. Infants born at 34–35 weeks of GA with HIE according to TH application.**

| Variables | Received TH (n = 31) | No TH (n = 15) | p |
|---|---|---|---|
| Gestational age (w)* | 34.9 ± 0.53 | 34.9 ± 0.54 | 0.749[w] |
| Birth weight (g)* | 2426 ± 421 | 2248 ± 481 | 0.206[y] |
| Resuscitation at DR‡ | 28 (90.3) | 14 (93.3) | 1.00[y] |
| Grade of HIE‡ | | | <0.001[y] |
| Mild (n = 12) | 1 (3.2) | 11 (73.3) | |
| Moderate (n = 19) | 18 (58.1) | 1 (6.7) | |
| Severe (n = 15) | 12 (38.7) | 3 (20) | |
| Underlying etiologies‡(**) | | | |
| Antepartum (maternal/fetal) | 15 (48.3) | 11 (73.3) | 0.11[y] |
| Intrapartum | 14 (45.1) | 6 (40) | 0.741[y] |
| Postpartum | 4 (12.9) | 1 (6.6) | 1.00[z] |
| Thompson scores§ | | | |
| At admission | 14.0 (10.0–17.0) | 5.0 (3.0–12.0) | 0.003[x] |
| $24^{th}$ h | 11.0 (6.0–15.0) | 3.0 (0.0–5.0) | <0.001[x] |
| $48^{th}$ h | 10.0 (4.0–14.0) | 1.0 (0.0–4.0) | <0.001[x] |
| $72^{nd}$ h | 7.0 (2.0–13.0) | 0.0 (0.0–2.0) | <0.001[x] |
| Accompanying organ system/condition ‡ | 28 (90.3) | 8 (53.3) | 0.008[y] |
| Abnormal EEG finding‡ | 9/18 (50) | 0/3 (0) | 0.229[y] |
| Abnormal MRI finding‡ | 13/16 (81.3) | 2/5 (40) | 0.115[y] |
| Mortality‡ | 9 (29) | 1 (6.7) | 0.132[y] |

Data given as: *mean ± SD, ‡number (%), §median (IQR); w:Student-t test, x:Mann Whitney U test, y:Chi-square test,z:Fisher-exact test.

DR: delivery room;. EEG: electroencephalography; HIE: hypoxic ischemic encephalopathy; MRI: magnetic resonance imaging.

## Discussion

This national population-based study revealed an incidence rate of 2.13 per 1000 live births in infants ≥36 weeks GA for HIE. This rate remained consistent over time when compared to previous national data [6], and the data from high-income countries [3]. However, mortality showed a downward trend (22.6% vs. 6%). In this study, we presented the care practices and outcomes of infants with HIE. We could also show data of infants with mild HIE and infants born at 34–35 weeks GA undergoing TH, for which there is still insufficient evidence in current practice.

Previous years' reports from different NICUs in our country showed that the rate of infants with HIE was between 3.5–6% in [22,23]. The incidence of HIE in infants ≥36 weeks GA accounted for 1.55% of all NICU admissions in this study. The decrease in incidence observed in this study might be attributed to the study's inclusion criteria, and the official Neonatal Resuscitation Program that has been widely implemented in Türkiye since the 2000s, with a focus on perinatal asphyxia prevention. Additionally, a national guideline established by Turkish Neonatal Society provides recommendations for both routine care of all newborns and resuscitation approaches for those requiring it [24].

Hypoxic-ischemic injury to the infant's brain may occur during antepartum and postnatal periods, but it occurs less often than during the intrapartum period. From 5% and 20% of neonatal HIE results from hypoxic ischemic damage in the antepartum period [21]. A large population-based observational study, indicates that 56% of all NE cases were related to hypoxic-ischemic injury that occurred during intrapartum period [25]. In developed countries, it is estimated that intrapartum hypoxic-ischemia is related to 30% of cases of NE, whereas in developing countries, it is related to 60% of cases [3]. Postpartum events, including cardiac failure and severe pulmonary disease, may alone cause HIE and could account for about 5% to

**Table 6. Findings of survivor and non-survivor infants.**

| Variables | Survivors (n = 907) | Non-survivors (n = 58) | p |
|---|---|---|---|
| Gestational age [*] | 38.9 ± 1.4 | 38.5 ± 1.5 | 0.044[w] |
| Birth weight [š] | 3223 ± 492 | 3166 ± 688 | 0.537[w] |
| Place of birth [‡] | | | <0.001[y] |
| Inborn | 485 (53.5) | 32 (55.2) | |
| Outborn | 420 (46.3) | 23 (39.7) | |
| Home | 2 (0.2) | 3 (5.2) | |
| APGAR scores [š] | | | |
| 1[st] min | 4.0 (3.0–5.0) | 2.0 (0.75–3.0) | <0.001[x] |
| 5[th] min | 6.0 (5.0–7.0) | 4.0 (3.0–5.0) | <0.001[x] |
| 10[th] min | 7.0 (6.0–8.0) | 5.0 (3.25–5.75) | <0.001[x] |
| 15[th] min | 7.0 (6.0–8.0) | 5.0 (5.0–7.0) | <0.001[x] |
| 20[th] min | 8.0 (7.0–9.0) | 6.0 (5.0–7.0) | <0.001[x] |
| Resuscitation at DR [š] | 749 (82.6) | 55 (94.8) | 0.011[z] |
| Cord blood gas analysis | | | |
| pH [š] | 6.98 (6.90–7.03) | 6.86 (6.80–7.02) | 0.140[w] |
| BE [š] | -16.0 [(-19.0)–(-13.0)] | -19.6 [(-24.12)–(-15.25)] | 0.011[x] |
| Blood gas analysis at 1[st] h [*] | | | |
| pH [š] | 7.09 (7.00–7.18) | 6.91 (6.75–7.03) | <0.001[w] |
| BE [š] | -14.0 [(-17.27)–(-10.0)] | -19.5 [(-22.6)–(-14.6)] | <0.001[x] |
| Thompson scores [š] | | | |
| Admission | 8.0 (5.0–12.0) | 16.0 (12.0–18.0) | <0.001[x] |
| 24[th] h | 5.0 (2.0–10.0) | 15.0 (12.0–17.25) | <0.001[x] |
| 48[th] h | 3.0 (0.0–7.0) | 15.0 (10.0–17.0) | <0.001[x] |
| 72[nd] h | 1.0 (0.0–5.0) | 14.0 (11.5–17.5) | <0.001[x] |
| Performed TH [‡] | 741 (81.7) | 58 (100) | <0.001[y] |
| Timing initiation of TH [š] | 2.0 (1.0–4.0) | 2.0 (1.0–4.0) | 0.794[x] |
| Clinical seizure at any time [‡] | 239 (26.4) | 31 (53.4) | <0.001[z] |
| Accompanying organ system/condition during TH [‡] | 572 (63.1) | 58 (100.0) | <0.001[z] |
| Abnormal EEG finding [‡] | 143/464 (30.8) | 3/3 (100.0) | <0.001[z] |
| Abnormal MRI finding [‡] | 207/627 (33.0) | 8/9 (88.9) | 0.001[z] |
| Length of hospital stay (d) [š] | 11.0 (8.0–16.0) | 3.5 (2.0–8.5) | <0.001[x] |

Data given as: [*]mean ± SD, [‡]number (%), [š]median (IQR); w:Student-t test, x:Mann Whitney U test, y:Chi-square test,z:Fisher-exact test.

DR: delivery room; EEG: electroencephalogram; MRI: magnetic resonance imaging TH: therapeutic hypothermia.

10% of cases. Assessing infants with HIE necessitates consideration of the history of pregnancy, labor, and delivery [19]. Based on our data, antepartum conditions (maternal/fetal) were responsible for 24.8% of cases, while intrapartum conditions accounted for 41.2%, and postpartum conditions in were found in 10.3% of infants. The remaining 23.7% of cases causes could not be definitively established, which is similar to the rate reported by Pierrat *et al.* [25]. According to the Vermont Oxford Network Encephalopathy Registry, up to 50% of NE cases have an unknown underlying etiology that cannot be attributed to asphyxia or inflammatory indicators [26].

The neurological symptoms that can occur with HIE aid in determining the presence, pattern and severity of the patient's hypoxic-ischemic injury. Standardization during neurological examination is accomplished using the APGAR score, modified Sarnat score and Thompson score [21]. The Sarnat scale classifies encephalopathy into three stages: mild (stage 1), moderate (stage 2) and severe encephalopathy (stage 3). The current recommendation advises starting TH for neonates with moderate to severe encephalopathy on the Sarnat assessment with historical and biochemical criteria [18]. The Thompson score including 9 independent clinical

items, is now increasingly used, considering its relevant predictive values for short-term outcomes and neurodevelopmental outcome at 24 months [27,28]. Thompson scores from admission to 3 days of life have predictive capability for death during hospitalization in this study.

Neurological dysfunction is only one aspect of the spectrum of injuries seen in NE resulting from hypoxia ischemic brain damage. Infants with HIE may also have concurrent multi-organ dysfunction, which increases the risk of morbidities and mortality [29]. In addition to having a compromised CNS, all infants with severe HIE exhibited signs of dysfunction in at least one additional organ or system dysfunction. Studies mostly reported involvement of pulmonary, hepatic, renal, and cardiovascular systems [15,30]. The following studies showed that the severity of multiorgan dysfunction was related to the severity of HIE [31,32]. In this study, it was observed that almost all infants with severe HIE had at least one accompanying organ system or condition with a higher incidence compared to other grades of HIE.

As per the current management guidelines, TH is recommended only for neonates ≥36 weeks gestational age with moderate to severe HIE [9,14]. The 2013 Cochrane review comprised 11 randomized controlled trials and involved 1505 infants. The findings of the review revealed that TH decreased the combined outcomes of mortality and major neurodevelopmental disability at 18 months, resulting in a decrease in mortality and reduced neurodevelopmental disability in survivors [9]. A considerable proportion of infants were reported to experience abnormal outcome during follow-up in also those with mild HIE [11]. For infants with mild HIE, there is currently insufficient data to recommend routine TH, yet significant advantages or risks cannot be ruled out [12]. A survey of cooling centers in the United Kingdom revealed that 75% of centers offered TH to infants with mild HIE [33]. In this study, almost all infants diagnosed with moderate/severe HIE received TH, and 34.6% of neonates with mild HIE received TH. Although long-term outcomes for infants with mild HIE are not available, the MRI and EEG findings of these infants were not different between those who received TH and who did not.

The current recommendation is to start TH within 6 h of birth [9,34,35]. Preclinical studies suggest that early initiation of TH improves neuroprotection [36,37]. However, a recent retrospective, observational, cohort study of Guillot *et al.* showed that early TH started before 3 h of life did not correlate with a reduction in brain lesions on MRI or better neurodevelopmental outcomes [38]. No significant differences in mortality and abnormal MRI findings were found in our study when evaluating the infants with moderate/severe HIE based on their TH initiation time (as <3 h, 3–6 h and >6 h). Although only 3.1% of infants received TH beyond 6 h (median 9.5 h), we compared the initiation time for TH within 6 h as <3 h and 3–6 h, and no significant difference was observed for mortality and abnormal MRI findings were observed. Long-term outcomes were unfortunately not included in this study.

The efficacy of TH in preterm infants remains uncertain. A study by Azzopardi *et al.* found that preterm infants born at 34 or 35 weeks of GA who received TH had a higher mortality rate compared to full-term infants [39]. The two small studies that evaluated the short- and long-term outcomes of TH in late preterm infants concluded that TH appears to be feasible in preterm infants. However, the incidence of complications and the combined outcome of death and neurodevelopmental outcomes in this vulnerable population is concerning [40,41]. The outcomes of a recent pilot study revealed that, at 24 months of age, late preterm infants with moderate HIE after TH demonstrated age-appropriate neurodevelopmental progress. However, the study strongly advised against using the data for clinical decision making [42]. In addition, the study reported a higher mortality rate in late preterm infants (34–35 weeks of GA) with HIE as compared to infants born ≥36 weeks of GA. The mortality rate was found to be similar in late preterm infants who received TH and those who did not. Our study, did not

find evidence that TH improved survival in this population, which could be attributed to small number of late preterm infants in our cohort or other unidentified factors leading to mortality.

Our findings suggest a mortality rate of 6% in this study, contrasting with the previously published national data which reported a rate of 22.6% [6]. According to HIE grading, there were no death in infants with mild HIE during both study periods. The current study reported mortality rates of 2.3% and 29.8% for infants with moderate and severe HIE, respectively; whereas previous data reported mortality rates of 6.6% and 51.7% for infants with moderate and severe HIE, respectively. We observed an improvement in mortality rates for moderate and especially of severe HIE. This discrepancy could be due to differences in distribution of HIE severity. While mild HIE incidences were comparable in two periods, there was a higher frequency of moderate HIE (58.9% vs. 38.7%), and a lower incidence of severe HIE in this period (15.6% vs. 31.2%). Vega-Del-Val *et al.* reported a temporal trend toward a decrease in severe HIE infants and a slight decline of mortality in a cross-sectional study conducted in Spain [43]. Most of the participating centers (81.2%) were the same in both periods, suggesting no major differences in the management of infants. The reduction of mortality is likely associated with the availability of TH in the new era and improved neonatal care throughout the country. In all cases, the severity of multiorgan dysfunction was thought to be the cause of death due to the lack of a legal basis for end-of-life decisions, which was the main reason for death in other series [18,44].

The use of nationwide data set from a database registry of prospectively recorded data, similar to those in other European countries [45–48], is the main strength of this study. This is the largest sample size available to determine the incidence of HIE in our country. This observational study conducted in tertiary NICUs, with attending neonatologists and pediatric neurologists, and facilities of servo-controlled TH, aEEG (mostly), conventional EEG and MRI. All participating NICUs followed the same diagnosis and staging criteria for HIE, and TH initiation. However, there are few limitations to our study. While 64 NICUs were made aware of the study, 42 (66%) of them chose to participate. We only included the number of infants with HIE who were born in participating centers and the number of deliveries that occurred there to determine the incidence of HIE. Outborn infants and the number of deliveries at sending hospitals were not included as some outborn infants were transferred to a participating center due to reasons such as not having a TH device, another patient receiving TH, or the NICU of the transferring hospital being at full capacity. This is not a randomized controlled trial, so there may be variations in management and interventions during TH. Approximately one-third of the study's infants did not have MRI results. Our data provided the short-term outcomes, but it is important to assess the long-term neurodevelopmental outcomes of infants.

## Conclusion

This study reveals a comparable HIE frequency and lower mortality rate compared to prior reports in our country. Early initiation of TH (<3h) did not demonstrate a decrease in brain lesions on MRI or mortality, however clinicians may attempt to initiate TH promptly after birth once the indication is confirmed. The registry analysis revealed that clinicians choose to administer TH to approximately one-third of infants with mild HIE and two-thirds of late preterm infants with HIE received TH, thereby indicating a broader application of TH in gray-zone patients. Due to the significant incidence of complications and the combined outcome of death and neurodevelopmental impairment, it is necessary to conduct longer follow-up studies in this vulnerable population.

## Supporting information

**S1 Checklist. STROBE statement—checklist of items that should be included in reports of observational studies.**
(DOCX)

## Author Contributions

**Conceptualization:** Emel Okulu, Ibrahim Murat Hirfanoglu, Mehmet Satar, Omer Erdeve.

**Data curation:** Emel Okulu, Mehmet Satar, Omer Erdeve, Esin Koc, Didem Armangil, Gaffari Tunc, Nihal Demirel, Sezin Unal, Ramazan Ozdemir, Melek Akar, Melike Kefeli Demirel, Merih Çetinkaya, Halime Sema Can Buker, Belma Saygılı Karagöl, Deniz Yaprak, Abdullah Barış Akcan, Ayse Anik, Fatma Narter, Sema Arayici, Egemen Yildirim, Ilke Mungan Akin, Ozlem Sahin, Ozgul Emel Bulut Ozdemir, Fahri Ovali, Mustafa Ali Akin, Yalcin Celik, Aysen Orman, Sinan Uslu, Hilal Ozkan, Nilgun Koksal, Ayhan Tastekin, Mehmet Gunduz, Ayse Engin Arisoy, Resat Gurpinar, Rahmi Ors, Huseyin Altunhan, Ramazan Kececi, Hacer Yapicioglu Yildizdas, Demet Terek, Mehmet Ates, Sebnem Kader, Mehmet Mutlu, Kıymet Çelik, Ebru Yucesoy, Mustafa Kurthan Mert, Selvi Gulasi, Kazım Kucuktasci, Didem Arman, Berna Hekimoglu, Nazlı Dilay Gultekin, Hasan Tolga Celik, Dilek Kahvecioglu, Can Akyildiz, Erdal Taşkın, Nukhet Aladag Ciftdemir, Saime Sundus Uygun, Tugba Barsan Kaya, Arzu Akdag, Aslan Yilmaz.

**Formal analysis:** Emel Okulu, Ibrahim Murat Hirfanoglu, Mehmet Satar, Omer Erdeve, Ferda Ozlu, Mahmut Gokce, Didem Armangil, Gaffari Tunc, Nihal Demirel, Sezin Unal, Ramazan Ozdemir, Mehmet Fatih Deveci, Melek Akar, Ilke Mungan Akin.

**Investigation:** Mehmet Satar, Ferda Ozlu, Mahmut Gokce.

**Methodology:** Emel Okulu, Ibrahim Murat Hirfanoglu, Mehmet Satar, Esin Koc, Mahmut Gokce.

**Project administration:** Ibrahim Murat Hirfanoglu, Mehmet Satar, Omer Erdeve, Esin Koc.

**Software:** Ferda Ozlu.

**Writing – original draft:** Emel Okulu.

**Writing – review & editing:** Mehmet Satar, Omer Erdeve, Esin Koc.

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
