## [Decision Letter · Decision Letter 0]

3 Oct 2023

PONE-D-23-19394AN OBSERVATIONAL, MULTICENTER, REGISTRY-BASED COHORT STUDY OF TURKISH NEONATAL SOCIETY IN NEONATES WITH HYPOXIC ISCHEMIC ENCEPHALOPATHYPLOS ONE

Dear Dr. Okulu,

Thank you for submitting your manuscript to PLOS ONE. After careful consideration, we feel that it has merit but does not fully meet PLOS ONE’s publication criteria as it currently stands. Therefore, we invite you to submit a revised version of the manuscript that addresses the points raised during the review process.

ACADEMIC EDITOR: Please go through all reviewers' comments and reply to point by point. We believe that your study is very interesting comparing data pre and post invention of therapeutic hypothermia.

We look forward to receiving your revised manuscript.

Kind regards,

Stefan Grosek, Ph.D., M.D.,

Academic Editor

PLOS ONE

Journal Requirements:

Additional Editor Comments:

Dear Authors

Two reviewers reviewed your article as myself and we found it interesting and well presenting. Please go through all comments and reply point by point.

Kind regards

Academic Editor

Reviewers' comments:

Reviewer's Responses to Questions

**Comments to the Author**

1. Is the manuscript technically sound, and do the data support the conclusions?

Reviewer #1: Yes

Reviewer #2: Yes

2. Has the statistical analysis been performed appropriately and rigorously? 

Reviewer #1: Yes

Reviewer #2: Yes

3. Have the authors made all data underlying the findings in their manuscript fully available?

Reviewer #1: Yes

Reviewer #2: Yes

4. Is the manuscript presented in an intelligible fashion and written in standard English?

Reviewer #1: Yes

Reviewer #2: No

5. Review Comments to the Author

Reviewer #1: The current study explains the prevalence of HIE in their own nation, where they found decline death rate as compare to the prior report. They concluded that early TH would be preferable because it was not linked to brain damage. They also came to the conclusion that the incidence of HIE stayed the same throughout time with previously published reports but the death rate decreased.

Please find the few Suggestions for improvement of the study

1- Please improve the quality of the figures. It's hard to read the figures.

2- In these HIE instances, were there any sex differences?

3- For more thorough information, also provide the sex details and their results should be supplied. Please discuss any outcome if available.

4-The long-term neurodevelopmental effects of newborns, which may be crucial to discuss before drawing any conclusions about when TH should start, are important aspects. Please modify the conclusion and discuss this in details.

Reviewer #2: The manuscript, which is under evaluation, is designed to deal with data from the multicentre,

observational, nationally based Register of Hypoxic Ischemic Encephalopathy (HIE).

The main strength of this study is to determine the incidence and progression of perinatal newborn

care during the analysed years based on a nationwide data set from a database registry of your country.

The added value of this study is to compare the treatment of newborns before and after the introduction of the national registry and, based on the finding’s improvement of treatment.

Congratulations to the authors for the introduced registry of newborns with HIE and the evaluation of their national data.

While reviewing this manuscript, I have a few comments, questions, and requests for minor corrections:

1. Minor corrections:

- line 366: number missing - value: "mild HIE: ......median"; please added.

- Line 508: need to be rewritten; a new term for HI "trauma" is needed?

2. Questions:

-- It is interesting that a postpartum event (probably in the first hours after birth), most likely defined

as Apparent Life-Threatening Event, occurred in 10.3% of newborns treated with hypothermia.

Do you have an explanation for this percentage of postpartum events underlying HIE?

- How can you explain findings in your study that only severe HIE is associated with

dysfunction of one organ system, when, we know that even mild and moderate hypoxia

is accompanied by impairment of organ systems; including the bone marrow.

- Before drawing conclusions about the differences between groups based on MRI and aEEG,

would it make sense to analyse their neurodevelopmental outcomes at 24 months of age?

Because HI is also a long-term process; when analysing the effectiveness of hypothermia,

we do not take into account the tertiary phase of HE brain damage,

which continues for a long time after the acute event. Do you have any data about

neurodevelopmental outcome of children treated with TH in newborn period?

- The same applies to the interpretation of TH in late preterm infants, please.

- Perhaps, a higher predictive value when comparing groups would be achieved if the

brain magnetic resonance imaging will be assessed based on

MRI scoring systems for HIE brain injury designed by Weeke and co-authors. I will suggest to use

Weekes MRI scoring system as a validated and standardized tool for defining HI brain injury

and as a marker for the analysis of two groups in your study.

- Your study highlights that the overall incidence of severe HIE has decreased; which is consistent

with the results of other studies. Given the fact that your study includes only inborn patients,

would the results be the same if you also included outborn neonates?

-With the introduction of the national registry of newborns with HIE, did you also notice a difference

in the treatment of such newborns before and after the introduction of the registry, such as achieving

the same cooling temperature between different centres; unified neuromonitoring,

unified imaging diagnostics and unified follow up?

Thank you,

6. PLOS authors have the option to publish the peer review history of their article (what does this mean?). If published, this will include your full peer review and any attached files.

Reviewer #1: **Yes: **Preeti Singh Chauhan

Reviewer #2: No

---

## [Author Response · Author response to Decision Letter 0]

16 Nov 2023

Dear Editor and Reviewers,

We appreciate the time you and the reviewers spent to review our paper and contribute valuable feedback. Your valuable and insightful comments inspired potential improvements to this current version. The authors have carefully considered the comments and have done our best to respond to each one. We hope that the manuscript will be able to live up to your high expectations following focused edits. Any further beneficial criticism will always be welcomed by the authors.

We have listed the point-to-point responses below, and added the manuscript with tracking changes. 

Sincerely,

Corresponding author

Emel Okulu

Response to Reviewers

Reviewer #1: 

Comment: Please improve the quality of the figures. It's hard to read the figures.

Response: The figures were revised and added. 

Comment: In these HIE instances, were there any sex differences?

Response: There were no sex differences according to HIE stages. This analysis is present in Table 1. 

Comment: For more thorough information, also provide the sex details and their results should be supplied. Please discuss any outcome if available.

Response: This information is present in results section under the heading ‘Data of infants ≥36 weeks GA according to grading of HIE’. The word ‘gender’ was changed to ‘sex distrubition’.

Comment: The long-term neurodevelopmental effects of newborns, which may be crucial to discuss before drawing any conclusions about when TH should start, are important aspects. Please modify the conclusion and discuss this in details.

Response: Long-term follow-up is essential for this population. The conclusion revised as you recommended. 

Reviewer #2: 

Comment: Minor corrections:- line 366: number missing - value: "mild HIE: ......median"; please added; - line 508: need to be rewritten; a new term for HI "trauma" is needed?

Response: The missing value added and ‘trauma’ changed as brain damage. 

Comment: It is interesting that a postpartum event (probably in the first hours after birth), most likely defined as Apparent Life-Threatening Event, occurred in 10.3% of newborns treated with hypothermia. Do you have an explanation for this percentage of postpartum events underlying HIE?

Response: This percentage may be high to due to meconium aspiration syndrome which was found as the most common cause for postpartum asphyxia.

Comment: How can you explain findings in your study that only severe HIE is associated with

dysfunction of one organ system, when, we know that even mild and moderate hypoxia

is accompanied by impairment of organ systems; including the bone marrow.

Response: In our study, it was found that it was not only the infants with severe HIE who had impairment of organ systems. Almost all infants (90.7%) with severe HIE, 71.8% of moderate HIE and 34.6% of mild HIE had at least one problem. This information is present in both text and Table 2. 

Comment: Before drawing conclusions about the differences between groups based on MRI and aEEG, would it make sense to analyse their neurodevelopmental outcomes at 24 months of age? Because HI is also a long-term process; when analysing the effectiveness of hypothermia, we do not take into account the tertiary phase of HE brain damage, which continues for a long time after the acute event. Do you have any data about neurodevelopmental outcome of children treated with TH in newborn period?

Response: Thank you for your comment, you are absolutely right. Unfortunately, we do not have the data about neurodevelopmental outcome of these children now, but we are planning another study to collect this data soon.

Comment: The same applies to the interpretation of TH in late preterm infants, please.

Response: We are planning another study to collect this data soon

Comment: Perhaps, a higher predictive value when comparing groups would be achieved if the brain magnetic resonance imaging will be assessed based on MRI scoring systems for HIE brain injury designed by Weeke and co-authors. I will suggest to use Weekes MRI scoring system as a validated and standardized tool for defining HI brain injury and as a marker for the analysis of two groups in your study.

Response: We thank for your recommendation. It has been shown that Weeke scoring system is a high importance to assess the impact of perinatal asphyxia on MR imaging. 

It would be good to use this scoring system. We reviewed our data in this respect, but we could not adjust our data on the extension of injury used for scoring. Therefore, we regret that we could not complete this suggestion. 

Comment: Your study highlights that the overall incidence of severe HIE has decreased; which is consistent with the results of other studies. Given the fact that your study includes only inborn patients, would the results be the same if you also included outborn neonates?

Response: The registry data contained information on the proportion of inborn and outborn infants at each participating center. Certain outborn infants were transferred to a participating center for one of the following three reasons: i. No TH device available, ii.The TH device is in use by another patient, iii. NICU capacity is full. It may be challenging and inaccurate to calculate the incidence by simply asking the number of deliveries from the sending hospital, so the only the incidence of HIE was calculated according to inborn HIE infants. 

Comment: With the introduction of the national registry of newborns with HIE, did you also notice a difference in the treatment of such newborns before and after the introduction of the registry, such as achieving the same cooling temperature between different centres; unified neuromonitoring, unified imaging diagnostics and unified follow up?

Response: Obviously we don’t know that. Nonetheless, the centers are maintaining more comprehensive rec

---

## [Decision Letter · Decision Letter 1]

29 Nov 2023

AN OBSERVATIONAL, MULTICENTER, REGISTRY-BASED COHORT STUDY OF TURKISH NEONATAL SOCIETY IN NEONATES WITH HYPOXIC ISCHEMIC ENCEPHALOPATHY

PONE-D-23-19394R1

Dear Dr. Okulu,

We’re pleased to inform you that your manuscript has been judged scientifically suitable for publication and will be formally accepted for publication once it meets all outstanding technical requirements.

Kind regards,

Stefan Grosek, Ph.D., M.D.,

Academic Editor

PLOS ONE

Additional Editor Comments (optional):

Dear Authors

I found that you addressed all issues I'll recommend to the Editor to accept your article for publication

Kind regards

Academic Editor

Reviewers' comments:

Reviewer's Responses to Questions

**Comments to the Author**

1. If the authors have adequately addressed your comments raised in a previous round of review and you feel that this manuscript is now acceptable for publication, you may indicate that here to bypass the “Comments to the Author” section, enter your conflict of interest statement in the “Confidential to Editor” section, and submit your "Accept" recommendation.

Reviewer #2: All comments have been addressed

2. Is the manuscript technically sound, and do the data support the conclusions?

Reviewer #2: Yes

3. Has the statistical analysis been performed appropriately and rigorously? 

Reviewer #2: I Don't Know

4. Have the authors made all data underlying the findings in their manuscript fully available?

Reviewer #2: Yes

5. Is the manuscript presented in an intelligible fashion and written in standard English?

Reviewer #2: Yes

6. Review Comments to the Author

Reviewer #2: Thank you for your answers and corrections.

The manuscript is processed for acceptance and publication in PLOS ONE journal.

7. PLOS authors have the option to publish the peer review history of their article (what does this mean?). If published, this will include your full peer review and any attached files.

Reviewer #2: No

---

## [Editor Report · Acceptance letter]

4 Dec 2023

PONE-D-23-19394R1 

An Observational, Multicenter, Registry-Based Cohort Study of Turkish Neonatal Society in Neonates with Hypoxic Ischemic Encephalopathy 

Dear Dr. Okulu:

I'm pleased to inform you that your manuscript has been deemed suitable for publication in PLOS ONE. Congratulations! Your manuscript is now with our production department. 

Kind regards, 

on behalf of

Professor Stefan Grosek 

Academic Editor

PLOS ONE